# AQUATIC-DIFF: ADDITIVE QUANTIZATION FOR TRULY TINY COMPRESSED DIFFUSION MODELS

## ABSTRACT

Tremendous investments have been made towards the commodification of diffusion models for generation of diverse media. Their mass-market adoption is however still hobbled by the intense hardware resource requirements of diffusion model inference. Model quantization strategies tailored specifically towards diffusion models have seen considerable success in easing this burden, yet without exception have explored only the Uniform Scalar Quantization (USQ) family of quantization methods. In contrast, Vector Quantization (VQ) methods, which operate on groups of multiple related weights as the basic unit of compression, have recently taken the parallel field of Large Language Model (LLM) quantization by storm. In this work, we for the first time apply codebook-based additive vector quantization algorithms to the problem of diffusion model compression, adapting prior works on the quantization-aware fine-tuning of transformer-based LLMs to take into account the special structure of convolutional weight tensors, the heterogeneity in the kinds of operations performed by the layers of a diffusion model, and the momentum-invalidating discontinuities encountered between successive batches during quantization-aware fine-tuning of diffusion models. We are rewarded with a data-free distillation framework which achieves to the best of our knowledge state-of-the-art results for the extremely low-bit weight quantization on the standard class-conditional benchmark of LDM-4 on ImageNet at 20 inference time steps. Notably, we report sFID 1.93 points lower than the full-precision model at W4A8, the best-reported results for FID, sFID and ISC at W2A8, and the first-ever successful quantization to W1.5A8 (less than 1.5 bits stored per weight) via a layer-wise heterogeneous quantization strategy. We thus establish a new Pareto frontier for diffusion model inference under low-memory conditions. Furthermore, our method allows for a dynamic trade-off between quantization-time GPU hours and inference-time savings, thus aligning with the recent trend of approaches that combine the best aspects of both Post-Training Quantization (PTQ) and Quantization-Aware Training (QAT). We are also able to demonstrate FLOPs savings on arbitrary hardware via an efficient inference kernel, as opposed to BOPs (Bit-wise Operations) savings resulting from small integer operations that may lack broad support across hardware of interest. Code is released via anonymized download link:
`https://osf.io/3uf8v/?view_only=ffbc957d6ce941d7b47bef09b628adcd`

## 1 INTRODUCTION

**Diffusion Models** (DM) (Ho et al., 2020; Dhariwal & Nichol, 2021; Rombach et al., 2021) have risen as the dominant architecture for many tasks, with a variety of established and emerging players investing in their commodification. The intense hardware resources involved in diffusion model inference have however proven a serious impediment. Bodies of work such as (Salimans & Ho, 2022; Meng et al., 2023) and (Song et al., 2021; Liu et al.; Lu et al., 2022) have seen outstanding success in reducing the number of model forward passes (denoising time steps) required for high-quality inference – down to as little as twenty steps, representing a fifty-fold reduction from (Ho et al., 2020). However, with one obstacle out of the way arises another, and with the latest-and-greatest open-source diffusion models such as SDXL 1.0 (Podell et al., 2024) boasting of 6.6 billion

parameters in total, the GPU VRAM and FLOPs requirements of a single forward pass are becoming a serious hindrance towards diffusion model inference on mass-market consumer hardware.

Fortunately, **model quantization** has emerged as a choice tool for radically shrinking generative models. Quantization methods balance the goal of lossy compression of model weights and activations to the maximum extent possible with the desire for minimal loss of generation quality. An impressive body of literature (Shang et al., 2023; Li et al., 2023; He et al., 2024b; Li et al., 2024; So et al., 2024; Wang et al., 2024; He et al., 2024a) has emerged on tailoring model quantization methods to the unique challenges posed by diffusion models. Historically these approaches have been split between Post-Training Quantization (PTQ) and Quantization-Aware Training (QAT). He et al. (2024a) have achieved excellent **W2A8** (two-bit weights and eight-bit activations) results on the class-conditional LDM-4 ImageNet model (Rombach et al., 2021) with an approach, that we denote *PTQ+PeFT*, melding the best aspects of PTQ and QAT through Parameter-Efficient Fine-Tuning.

Despite these successes, **substantial holes** exist in the model quantization literature on diffusion models. In contrast to the codebook-based Vector Quantization (VQ) approaches such as (Tseng et al., 2024; Egiazarian et al., 2024) that have come to dominate the Pareto frontier of Large Language Model (LLM) quantization, all works on diffusion model quantization to date have focused on Uniform Scalar Quantization (USQ)-based approaches. Furthermore, while LLM quantization to as little as a single bit per original weight (Xu et al., 2024) has been achieved, there has been no successful binarization of a large class-conditional latent diffusion model to date.

**In this paper**, we tackle for the first time the question of whether the codebook-based VQ approaches are also applicable to diffusion models, whose convolutional U-Net architecture (Ronneberger et al., 2015) and iterative denoising process has no analogue in the NLP domain. In the process, we uncover many surprising results. Starting with the framework of layer-by-layer independent calibration followed by whole-model parameter-efficient fine-tuning, we identify several issues, such as the unsuitability of AdamW for fine-tuning of diffusion models quantized with a learnt codebook and the non-independence of successive minibatches encountered along the denoising trajectory. We introduce a solution in

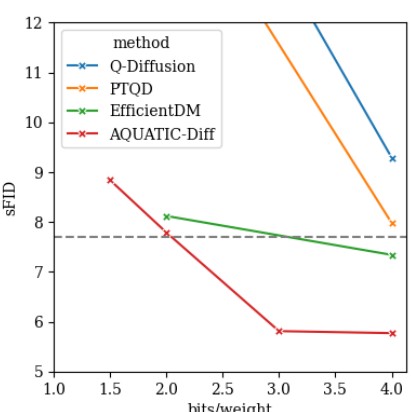

Figure 1: An illustration of the sFID (Nash et al., 2021) of our method on LDM-4 ImageNet at a variety of weight quantization levels, versus earlier approaches. The gray dashed line indicates the original model performance. Our sFID superiority at every bit-width establishes a new Pareto frontier.

the form of Selective Momentum Invalidation PV-Tuning (SeMI-PV). Furthermore, we observe and test opportunities for optimization in the form of Convolutional Kernel-Aware Quantization (KAQ) and Layer Heterogeneity-Aware Quantization (LAQ) for further weight savings.

Importantly, we contribute a complete, data-free and rapid PTQ+PeFT solution for the learnt codebook-based additive quantization of DMs that achieves **outstanding results** on the commonly-accepted metrics of Inception Score (IS) (Salimans et al., 2016), Fréchet Inception Distance (FID) (Heusel et al., 2017) and sFID (Nash et al., 2021), as shown in Fig. 1. At **W4A8**, our quantized model achieves FID and sFID that are both *better*, respectively by 1.75 and 1.93 points, than that of the non-quantized model on the standard ImageNet task with LDM-4, thus strongly indicating that even outside of any resource concerns, it is always better to use our quantized model over the original model. At **W2A8** on the same task, our FID, sFID and IS are respectively 1.13, 0.33 and 38.41 points better than the best existing solution of He et al. (2024a). Furthermore, via heterogenous quantization of different kinds of U-Net layer, we achieve an unprecedented **W1.5A8**, a 95.3% compression of the original weights. As He et al. (2024a) raised the importance of rapid and efficient quantization, our technique permits a trade-off between quantization cost and inference-time performance, with our most time-consuming stage being highly parallelizable. Due to the concerns of latency in addition to VRAM usage (our optimization focus), we show that our approach is the first to permit FLOPs reduction on arbitrary hardware, whereas prior approaches focus on savings enabled by hardware support for very low-bit integer operations that may not be universal.

## 2 BACKGROUND AND RELATED WORK

### 2.1 DIFFUSION MODELS

Diffusion models (Ho et al., 2020; Song et al., 2021) are a class of latent-variable generative model inspired by non-equilibrium thermodynamics, notable for the iterative forward and reverse processes by which they relate the data distribution to an isotropic Gaussian. In the basic case, the forward process is a Markov chain which repeatedly adds Gaussian noise to the sample:

$$q(\vec{x}_t|\vec{x}_{t-1}) = \mathcal{N}(\vec{x}_t; \sqrt{1-\beta_t}\vec{x}_{t-1}, \beta_t \mathbf{I}) \tag{1}$$

where the variance schedule $\beta_t \in (0,1)$ controls the amount of noise added in each of $T$ time steps. The reverse process is then approximated by a learned conditional distribution:

$$p_\theta(\vec{x}_{t-1}|\vec{x}_t) = \mathcal{N}(\vec{x}_{t-1}; \tilde{\vec{\mu}}_{\theta,t}(\vec{x}_t), \tilde{\beta}_t \mathbf{I}). \tag{2}$$

where at each denoising time-step $\tilde{\vec{\mu}}_{\theta,t}(\vec{x}_t)$ is calculated by a noise estimation network with shared weights. Model quantization induces error in the value of $\tilde{\vec{\mu}}_{\theta,t}(\vec{x}_t)$ at each time-step.

The cost of diffusion model inference is subsequently determined by the number of time steps at which noise prediction must be carried out as well as the cost of model inference for a single instance of noise prediction. Accelerated sampling strategies such as the DDIM Song et al. (2021), PLMS sampler Liu et al. and DPM-Solver Lu et al. (2022) seek to reduce the number of denoising time steps, whereas quantization approaches, such as our solution, target the cost of noise prediction.

### 2.2 DIFFUSION MODEL QUANTIZATION

Earlier works on the quantization of diffusion models, such as PTQ4DM (Shang et al., 2023), Q-Diffusion (Li et al., 2023), PTQD He et al. (2024b), Q-DM (Li et al., 2024) and TDQ (So et al., 2024) have noted a distinction between PTQ and QAT. QAT approaches are characterised by a costly fine-tuning process akin to knowledge distillation and/or access to the original training dataset, whereas PTQ involves the relatively lightweight layer-wise optimization of quantization parameters.

More recently, however, works such as EfficientDM (He et al., 2024a) and QuEST (Wang et al., 2024) have introduced a concept we label PTQ+PeFT, involving layer-wise alignment followed by parameter-efficient fine-tuning. Such approaches achieve inference-time results matching those of QAT, but are closer to PTQ in terms of resources required at quantization time. They thus combine the best aspects of both approaches.

### 2.3 QUANTIZATION STRATEGIES

Previous works on the quantization of diffusion models such as (Li et al., 2023) have exclusively focused on USQ (Fig. 2), where each weight is individually mapped from its full-precision floating-point representation $w$ to a low-bit integer $\hat{w}$ via a learnt affine transformation:

$$\hat{w} = \text{s} \cdot \text{clip}(\text{round}(\frac{w}{s} - z), c_{\min}, c_{\max}) + z, \tag{3}$$

where $c_{\min}$ and $c_{\max}$ are the smallest and largest integer representable at the chosen bit-width and $s, z$ are the learnt layer-wise or channel-wise scale factor and zero-point by which the transformation is parameterised. Works such as So et al. (2024); He et al. (2024a) have improved the flexibility of USQ by learning separate quantization parameters at each time-step.

Meanwhile, in the parallel field of LLM quantization, recent state-of-the-art works such as QuIP# (Tseng et al., 2024) and AQLM (Egiazarian et al., 2024; Malinovskii et al., 2024) have achieved impressive results with *Vector Quantization* (VQ) of model weights. Under $k$-bit vector quantization with $M$ codebooks, groups of $d$ weights each are jointly replaced with $M$ indices or codes $\in \mathbb{Z}_{kd/M}$ into codebooks $C^{(1)}, \ldots, C^{(M)} \in \mathbb{R}^{2^{kd/M} \times d}$. We extend this approach to diffusion models (Fig. 3).

### 2.4 ADDITIVE QUANTIZATION

**AQLM** (Egiazarian et al., 2024) introduced the use of *Additive Quantization* (AQ) as its vector quantization method (Figure 3), whereby each group of weights is reconstituted as the sum of its

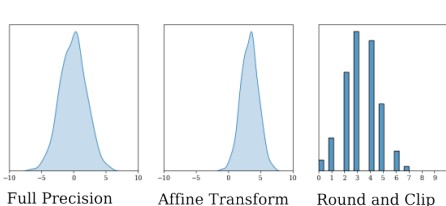

Figure 2: The Uniform Scalar Quantization (USQ) strategy.

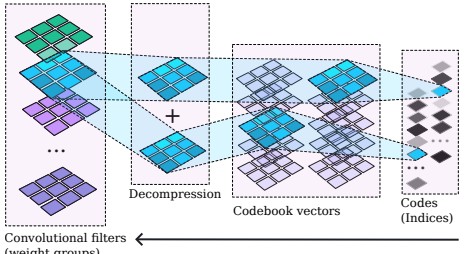

Figure 3: Additive Quantization (AQ) (Egiazarian et al., 2024), applied to a convolutional kernel.

indexed codebook vectors according to the following equation:

$$\widehat{\mathbf{W}} = \sum_{m=1}^{M} C_{b_{1,m}}^{(m)} \oplus \cdots \oplus \sum_{m=1}^{M} C_{b_{2^{kg/M},m}}^{(m)}, \tag{4}$$

with $\oplus$ as the concatenation operator and $b_{im} \in \mathbb{R}^{2^{kg/M}}$ as the code assigned to the $i$-th group of weights and $m$-th codebook under $k$-bit quantization, where $g$ is the group size and $M$ the number of codebooks. Quantization in Egiazarian et al. (2024) is carried out primarily in successive layer-by-layer fashion. The codes and codebooks for the layer are optimized in alternating fashion to minimize $||\mathbf{WA} - \widehat{\mathbf{W}}\mathbf{A}||_2^2$ on calibration data, with code optimisation carried out via beam search and codebook quantization carried out via Adam Kingma & Ba (2015). Subsequent Adam optimisation of all codebooks simultaneously is suggested as a whole-model PEFT solution and in this scenario codes are kept frozen. Malinovskii et al. (2024) instead develop the *PV-Tuning* algorithm for joint optimisation of both codes and codebooks against an arbitrary loss on a whole-model basis. Readers are directed to consult Egiazarian et al. (2024); Malinovskii et al. (2024).

The three important hyperparameters which determine the achieved bit-width under AQLM are the number of codebooks $M$, the group size $g$ and the size of the codebook indices, which we may fix as $n = kg/M$ for $k$-bit weight quantization. Note that there is some contribution to bit-width from the size of the code-book itself. $n = 8$ results in a small codebook of only 256 rows.

## 3 VECTOR QUANTIZATION OF DIFFUSION MODELS

Recent works on codebook-based vector quantization of generative models (Tseng et al., 2024; Egiazarian et al., 2024; Malinovskii et al., 2024) have focused on transformer-based LLMs and the quantization of fully-connected or linear layers. Diffusion models differ from LLMs in several key aspects, including the iterative denoising procedure by which they produce a sample and also the U-Net architecture, which features $3 \times 3$ and $1 \times 1$ convolutions in addition to linear layers. In the following sections, we illustrate the novel modifications we make to harmonize earlier vector quantization and diffusion model quantization approaches in light of these challenges.

Our approach operates as a two-step process. In the first stage, we convert each layer of the model to a vector-quantized layer, by means of per-layer calibration according to the procedure described in Egiazarian et al. (2024), so as to minimize a calibration loss $\arg\min||\mathbf{WA} - \widehat{\mathbf{W}}\mathbf{A}||_2^2$ for each layer independently. In the second stage, we perform parameter-efficient fine-tuning using the optimizer of Malinovskii et al. (2024), so as to minimise a teacher-student loss (Section 3.2).

### 3.1 STAGE 1: LAYER-BY-LAYER CALIBRATION

**Layer-Wise Calibration.** The AQLM algorithm as presented in Egiazarian et al. (2024) can only be applied via one-layer-at-a-time calibration on a finite (small) calibration dataset. Calibration of a layer is performed for a number of epochs until an early stopping criterion is met. The calibration dataset is generated via uniform random sampling at all time steps, as described in Li et al. (2023). The number of calibration images used and related hyperparameters are elucidated in Section 4.1.

**Additive Quantization of Convolutional Layers.** Egiazarian et al. (2024) only describes the AQLM compressed weight format in terms of fully-connected layers. However, we may easily extend it to convolutional layers of arbitrary stride, padding and kernel size, by noting that a $k$-strided $p$-padded $n \times n$ convolution may be exactly represented as a $k$-strided $p$-padded *sliding window view* of the input, followed by matrix multiplication with a rearrangement of the weights tensor. These re-indexing operations are completely transparent to automatic differentiation.

**Convolutional Kernel-Aware Quantization (KAQ).** Consider a convolutional layer with a weights matrix $F$ comprised of $C_{out}$ individual $C_{in} \times h_1 \times w_1$ filters $\{F_i\}_{i=1}^{C_{out}}$. The forward pass against an input $H$ may be expressed as the channel-wise concatenation

$$G = \bigotimes_{i=1}^{C_{out}} H * F_i, \qquad (5)$$

where $F \in \mathbb{R}^{C_{out} \times C_{in} \times h_1 \times w_1}$, $F_i \in \mathbb{R}^{C_{in} \times h_1 \times w_1}$, $H \in \mathbb{R}^{C_{in} \times h \times w}$, $H * F_i \in \mathbb{R}^{h \times w}$, and $*$ is the convolution operator. Due to the resulting correlation between weights corresponding to the same input or output channel, earlier works on diffusion model quantization such as (Li et al., 2023; Wang et al., 2024; Huang et al., 2024; He et al., 2024a) have all chosen to learn $C_{in}$ separate scales $s \in \mathbb{R}^{C_{in}}$ for the weight quantization according to (Equation 3). Meanwhile, the VQ approach Egiazarian et al. (2024) chooses to apply per-output-feature scaling subsequent to the quantized matmul operation $Y = X\widehat{W} * s$ corresponding to $s \in \mathbb{R}^{C_{out}}$. These scaling operations are illustrated in Fig. 4.

In either case, it is not possible to learn scale factors corresponding to both the input and the output channel dimension, as the prohibitive $C_{in} * C_{out}$ number of scale factors required would erase any gains from weight quantization. However, specifically in the case of additive vector quantization applied to $3 \times 3$ convolutional kernels, we may still achieve independent quantization of each individual $3 \times 3$ filter matrix corresponding to one input and one output channel, via considering each such matrix as the group of 9 weights to be replaced as a unit by one index per codebook. This choice is illustrated in Fig. 5.

Empirically, we observe a small improvement in the overall FID, sFID and ISC when the group size for additive quantization is set to exactly nine, as shown in our ablation study.

**Layer Heterogeneity-Aware Quantization (LAQ).** Diffusion models are deep neural networks, containing hundreds of layers. Furthermore, these layers vary in the type of operation performed, involving not only $3 \times 3$ convolution layers, but also $1 \times 1$ point-wise convolutional layers involved in attention operations, as well as linear layers involved in time embedding. As shown by Fig. 6, there is a trade-off unique to each kind of layer between the overall quantization error and the average number of bits used to store each parameter of the layer. $1 \times 1$ convolutional layers contribute to total model MSE more than $3 \times 3$ convolutional layers while constituting a small proportion of the total model parameters. Meanwhile, earlier works such as Huang et al. (2024); So et al. (2024); Wang et al. (2024) have found that accurately maintaining temporal information (encoded in the fully-connected layers) is especially important for high-quality image generation. We thus choose in the most extreme case to quantize

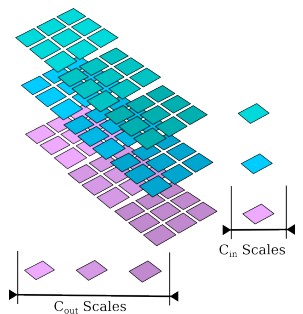

Figure 4: Scale factors.

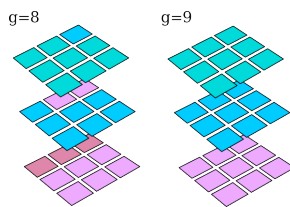

Figure 5: Effects of group size.

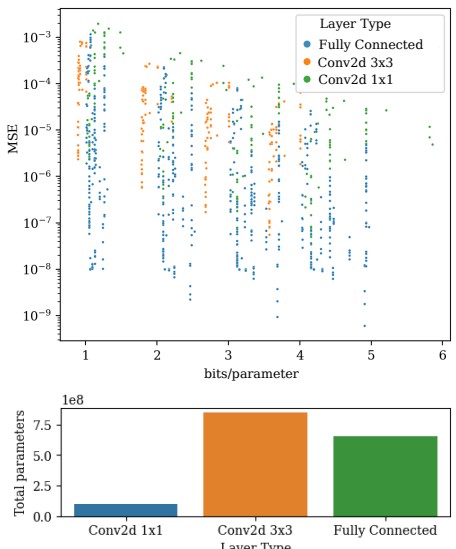

Figure 6: Top: Overall contribution to the quantization error versus the number of bits used per parameter, for all layers of LDM-4 ImageNet (number of codebooks $\in [1, 4]$). Bottom: Total count of parameters by layer.

$3 \times 3$ convolutional layers using one codebook per layer, while using two codebooks each for other layers, enabling our unprecedented **W1.5A8** result on the ImageNet LDM-4 model.

## 3.2 STAGE 2: PARAMETER-EFFICIENT FINE-TUNING

Subsequent to the Layer-Wise Calibration, PTQ+PeFT works such as (He et al., 2024a; Wang et al., 2024) additionally perform parameter-efficient fine tuning on a whole-model basis in data-free teacher-student knowledge-distillation fashion. The full-precision model is used to generate a batch of sample images from noise for a total of $T$ time-steps. At each denoising time-step $0 < t \leq T$, noise prediction is conducted via both the original model (the teacher) and the quantized model (the student). Then, the teacher-student loss

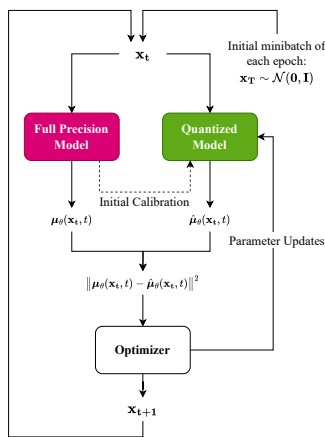

$$L_t = \left\| \boldsymbol{\mu}_\theta(\mathbf{x_t}, t) - \hat{\boldsymbol{\mu}}_\theta(\mathbf{x_t}, t) \right\|^2, \quad (6)$$

where $\boldsymbol{\mu}_\theta(\mathbf{x_t}, t)$ is the full-precision model and $\hat{\boldsymbol{\mu}}_\theta(\mathbf{x_t}, t)$ the quantized model, is computed and the optimizer advanced by one step. One epoch and $T$ optimization steps of PeFT thus correspond exactly to the generation of one batch of images via $T$ denoising time-steps.

Figure 7: The fine-tuning process.

**Discrete Optimisation using PV-Tuning.** The standard AdamW optimizer (Loshchilov & Hutter, 2019) can only perform continuous optimization of the learnt codebook vectors used for additive vector quantization, as opposed to discrete optimization of the learnt codebook indices used to represent each group of weights. In practice, we find optimization using AdamW to produce poor results (Section 4.4). We instead opt for the PV-Tuning optimizer of Malinovskii et al. (2024), which performs both continuous and discrete optimization.

**Selective Momentum Invalidation PV-Tuning (SeMI-PV).** With the outlined fine-tuning approach, we note divergence at the start of each denoising epoch (Fig. 8). We posit this to be due to the successive denoising process violating standard assumptions. Specifically, at the end of one epoch and the beginning of another, a batch of almost-fully denoised images is immediately followed by a batch of pure isotropic Gaussian noise. As a result the accumulated momentum is no longer valid. We solve this by simply resetting the optimizer state at the end of each epoch. As this is observed in our ablation to

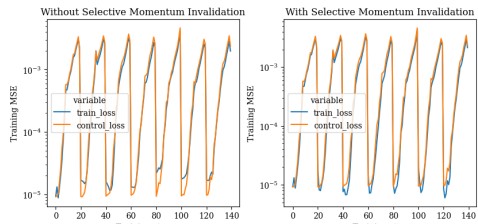

Figure 8: The convergence of PV-Tuning with versus without SeMI-PV.

result in effective training when instituted along with PV-Tuning instead of Adam, we dub the combination Selective Momentum Invalidation PV-Tuning (SeMI-PV).

**Adaptable Training-Time Denoising Schedule.** Earlier works on PTQ+PeFT approaches (He et al., 2024a; Wang et al., 2024) have placed importance on the time required for the quantization process in addition to the inference-time generation quality of the quantized model. While our focus is on state-of-the-art results for the compression of model weights, we would also like our method to be quite fast to perform. To this end, we note that unlike earlier works, the number of denoising steps used per epoch during the parameter-efficient fine-tuning is adaptable and may be reduced for accelerated training with minimal cost to inference-time performance (Section 4.4).

## 3.3 INFERENCE-TIME FLOPS REDUCTION WITH LUT MULTIPLICATION

$3 \times 3$ convolutional operations represent the majority of the inference-time FLOPs requirement for inference of the diffusion model U-Net. For any given such convolutional layer quantized using $k$-bit additive quantization with $M$ codebooks of size $\mathbb{R}^{2^{9k/m} \times 9}$, in cases where $C_{in} > M \cdot 2^{9k/m}$ a substantial reduction in FLOPs may be achieved on arbitrary hardware via an efficient inference kernel which precomputes the product of every codebook vector with each input patch prior to

dequantization. Empirical FLOPs are stated in Section 4.5, in contrast to earlier works such as (Li et al., 2023), which provides only theoretical computations of BOPs (Bitwise OPerations) assuming capability of the hardware to efficiently perform very low-precision integer operations. Detailed specifications of this kernel and proof of FLOPs reduction are provided in the Appendix.

Note that an analogous technique is used in the released code of Egiazarian et al. (2024) to accelerate the inference of fully-connected layers. Our contribution is in highlighting its applicability.

# 4 EXPERIMENTS

## 4.1 IMPLEMENTATION DETAILS

**Evaluation Methodology.** In order to demonstrate the general applicability of our methods, we evaluate our proposed technique on two widely-adopted benchmarks: Unconditional generation using the DDIM model of Song et al. (2021) on CIFAR-10 $64 \times 64$, and conditional generation using the LDM-4 model of Rombach et al. (2021) on ImageNet $256 \times 256$ (Deng et al., 2009). We compare the generation quality of our model with that of previous works using Inception Score (IS), Fréchet Inception Distance (FID) (Heusel et al., 2017), Spatial FID (sFID) (Salimans et al., 2016) and Precision. Inception Score is known to be unreliable on datasets other than ImageNet. However, we report it for CIFAR-10 so as to enable apples-to-apples comparison with earlier works on quantization. All metric calculations are conducted using the reference implementation from the ADM evaluation suite (Dhariwal & Nichol, 2021) after generation of 50,000 images via the quantized model. FLOPs are calculated using FAIR's *fvcore* (AI, 2019) tools and PyTorch compiled with MKL.

For our reporting of bit-width, it should be noted that weight quantization using VQ-based methods results in a decimal value for the average parameters/weight. For reasons of symmetry with all earlier works on diffusion model quantization, we conservatively round up to the nearest 0.5.

**Calibration Technique.** Uniform sampling of model inputs at all inference time steps is performed as in (Li et al., 2023), resulting in a calibration dataset of 5120 model inputs. Layer-by-layer weight quantization is subsequently carried out via AQLM (Egiazarian et al., 2024) with early-stopping at a relative error tolerance of 0.01. In line with earlier works such as (Li et al., 2023; Huang et al., 2024; So et al., 2024; He et al., 2024a; Wang et al., 2024), only the U-Net of latent diffusion models (LDMs) is quantized. The encoder and decoder which produce the latent representation are not quantized. Furthermore, the first and last convolutional layers of U-Nets are not quantized, due to their extremely small share of the parameter count and model FLOPs. The number of codebooks for AQLM quantization is set to $M = 4$ for **W4A8**, $M = 3$ for **W3A8**, and $M = 2$ for **W2A8**. For **W1.5A8**, $3 \times 3$ convolutional layers are quantized with $M = 1$ and all others layers $M = 2$. A group size of $d = 9$ is used for $3 \times 3$ convolutional layers and $d = 8$ for all other layers. The codebook size is set to $2^8$ entries per codebook, corresponding to 8-bit indices. This ensures that codebooks are relatively small compared to quantized weight matrices.

Modifications to these settings are noted in the subsection specific to the experiment.

**PeFT Hyperparameters.** Unless explicitly noted otherwise, whole-model PeFT is subsequently carried out using the PV-Tuning optimizer (Malinovskii et al., 2024) for 160 epochs of 100 successive denoising steps each in the W4A8 and W3A8 case and 320 epochs of 100 successive denoising steps each in the W2A8 and W1.5A8 cases, with a continuous optimization learning rate of $4e - 5$ decaying linearly to $1e - 6$ and a discrete optimization learning rate of $1e - 4$. A batch size of 4 is used for PeFT of LDM models and 64 for DDIM models.

**Activation Quantization Methodology.** Improvements to the quantization of weights, not activations is the focus of this paper. Consequently we quantize activations according to the methodology of (Li et al., 2024) for CIFAR-10 WxA8. For ImageNet, we use separate activation scale factors for each time-step as in He et al. (2024a); Wang et al. (2024), due to the well-attested better performance. Activation quantization is performed as the last step after PeFT has completed.

## 4.2 Unconditional Generation via DDIM CIFAR-10 $32 \times 32$

In line with earlier literature, we perform unconditional image generation on the CIFAR-10 dataset (Krizhevsky, 2009) at the W4A8 quantization level using the DDIM model of Song et al. (2021) (Table 1). We train on a single RTX3090 and perform inference at 100 denoising time steps, with $eta = 0.0$ and $cfg = 3.0$, in line with earlier works. We test against PTQ (Shang et al., 2023; Li et al., 2023), QAT (Esser et al., 2020; So et al., 2024) and PTQ+PeFT (He et al., 2024a) methods. On the balance of it, we outperform all other methods with regards to FID, with the exception of the QAT method TDQ (So et al., 2024) and the PTQ+PeFT method EfficientDM (He et al., 2024a). However, the source code of TDQ (So et al., 2024) is not available, and the released code of (He et al., 2024a) does not include the CIFAR-10 experiments. We have been unable to independently replicate their results. Excluding TDQ and EfficientDM, AQUATIC-Diff is best-in-class on the DDIM CIFAR-10 task. Furthermore, our GPU time requirements for quantization are much closer to that of PTQ than that of QAT, in line with our status as a PTQ+PeFT method.

Table 1: Performance comparison of our method on DDIM CIFAR-10 $32 \times 32$.

| Method | Bit-width (W/A) | Training data | GPU Time (hours) | Model Size (MB) | IS↑ | ID↓ |
|---|---|---|---|---|---|---|
| FP | 32/32 | 50K | - | 136.4 | 9.12 | 4.14 |
| PTQ4DM | 4/8 | 0 | 0.95 | 17.22 | 9.31 | 10.12 |
| Q-Diffusion | 4/8 | 0 | 0.95 | 17.22 | 9.12 | 4.93 |
| LSQ | 4/8 | 50K | 13.89 | 17.22 | 9.38 | 4.53 |
| TDQ | 4/8 | 50K | 16.99 | 17.26 | **9.59** | 4.13 |
| EfficientDM | 4/8 | 0 | 0.97 | 17.26 | 9.41 | **3.80** |
| AQUATIC-Diff | 4/8 | 0 | 3.66 | 17.35 | 9.00 | 4.43 |

Note that although our FID is superior to that of Shang et al. (2023); Li et al. (2023); Esser et al. (2020), our IS is substantially lower. Li et al. (2023) indicates that "[...] IS is not an accurate reference for datasets that differ significantly from ImageNet's domain and categories."

### 4.2.1 Conditional Generation via LDM-4 ImageNet $256 \times 256$

The highlight of our work is conditional image generation on the ImageNet (Deng et al., 2009) dataset at the W4A8 quantization level using the LDM-4 model of Rombach et al. (2021). We perform inference at 20 denoising time steps via the DDIM sampler of Song et al. (2021), with $eta = 0.0$ and $cfg = 3.0$, in line with earlier works. We test against all three applicable earlier works: Li et al. (2023), He et al. (2024b), and He et al. (2024a). Our results are displayed in Table 2.

We achieve impressive results across the board. At the **W4A8** level of quantization, we achieve FID and sFID that respectively outperform the full-precision model by 1.75 and 1.93 points. Furthermore, we exceed the best existing solution of (He et al., 2024a) by 1.57 points of sFID. At the **W3A8** level of quantization, our FID is 3.44 points better than that of the original model. At the **W2A8** level of quantization, our FID is 1.13 points lower, IS 39.2 points higher and precision 13.49 percentage points higher than that of (He et al., 2024a). Lastly, our novel **W1.5A8** level of quantization, where each weight is quantized with only 1.5 bits on average, results in FID that is still 2.3 points better than the full-precision model.

**Pareto Optimality.** This result establishes us as the *Pareto frontier* for this task, since our solution is the optimal choice for generation quality at every level of weight compression.

### 4.3 Quantization Efficiency

While our focus in the previous section is the achievement of the best possible FID and sFID at a given level of quantization, some earlier works such as EfficientDM (He et al., 2024a) have also stressed the importance of GPU resources required for quantization. We see this concern primarily as one of *wall-clock time*, and not necessarily of GPU hours, as even two days of time on an NVIDIA RTX 3090 would pessimistically cost just 12 dollars at prevailing market rates – a minuscule amount relative to the throughput of user requests seen by a commercial service such as DALLE-2 or Midjourney. Consequently, we would like to note that:

Table 2: Performance comparison of our method on LDM-4 ImageNet $256 \times 256$.

| Method | Bit-width (W/A) | IS↑ | FID↓ | sFID↓ | Precision↑ (%) |
|---|---|---|---|---|---|
| FP | 32/32 | 364.73 | 11.28 | 7.70 | 93.66 |
| Q-Diffusion | 4/8 | 336.80 | 9.29 | 9.29 | 91.06 |
| PTQD | 4/8 | 344.72 | **8.74** | 7.98 | 91.69 |
| EfficientDM | 4/8 | 353.83 | 9.93 | **7.34** | 93.10 |
| AQUATIC-Diff | 4/8 | **356.18** | 9.53 | **5.77** | **93.33** |
| AQUATIC-Diff | 3/8 | **333.57** | 7.84 | 5.81 | 92.39 |
| Q-Diffusion | 2/8 | 49.08 | 43.36 | 17.15 | 43.18 |
| PTQD | 2/8 | 53.36 | 39.37 | 15.14 | 45.89 |
| EfficientDM | 2/8 | 175.03 | 7.60 | 8.12 | 78.90 |
| AQUATIC-Diff | 2/8 | **213.44** | **6.47** | **7.79** | **92.39** |
| AQUATIC-Diff | **1.5/8** | **174.24** | **8.98** | **8.84** | **78.08** |

- Our Layer-by-Layer Calibration process is *embarassingly parallel*, that is to say, it may be split across 4 RTX 3090 GPUs for a $4\times$ speedup.

- Unlike approaches such as He et al. (2024a); Wang et al. (2024) which require quantization-aware fine-tuning of an LDM-4 ImageNet model at 100 time steps prior to inference at 20 time-steps, we are able to conduct both the PeFT process and inference at 20 time-steps.

Under the above optimizations, we observe only a small degradation of generation quality while maintaining comparable wall-clock time to He et al. (2024a) (Table 3).

Table 3: Efficiency comparison of our method on LDM-4 ImageNet $256 \times 256$.

| Method | Bit-width (W/A) | Wall Time (hours) | IS↑ | FID↓ | sFID↓ | Precision↑ (%) |
|---|---|---|---|---|---|---|
| FP | 32/32 | N/A | 364.73 | 11.28 | 7.70 | 93.66 |
| EfficientDM | 4/8 | **3.05** | **353.83** | 9.93 | **7.34** | **93.10** |
| AQUATIC-Diff | 4/8 | 5.61 | 350.28 | **9.04** | **5.77** | 92.80 |
| EfficientDM | 2/8 | **3.11** | 175.03 | **7.60** | **8.12** | **78.90** |
| AQUATIC-Diff | 2/8 | 6.95 | **180.57** | 8.10 | 9.58 | 78.87 |

## 4.4 ABLATION STUDY

We comprehensively ablate the considerations mentioned in Section 3 at the **W2A8** quantization level on the LDM-4 ImageNet model at 100 PeFT time-steps and 20 inference time-steps (Table 4). First, we set the group size $g = 9$ according to KAQ, and trial PeFT methods against the control of only layer-wise calibration, settling on our final **W2A8** method with SeMI-PV. Then, we see the effect of setting $g = 8$ instead, which is a small decrease in performance. Lastly, we apply LAQ and use only one codebook per $3 \times 3$ convolutional layer, thereby achieving **W1.5A8**.

Table 4: Ablation of our method on LDM-4 ImageNet $256 \times 256$.

| Method | Bit-width (W/A) | IS↑ | FID↓ | sFID↓ | Precision↑ (%) |
|---|---|---|---|---|---|
| FP | 32/32 | 364.73 | 11.28 | 7.70 | 93.66 |
| Layer-Wise Calibration + KAQ | 2/8 | 12.73 | 130.78 | 41.71 | 15.37 |
| + PeFT (AdamW) | 2/8 | **229.89** | **6.16** | **7.08** | **87.49** |
| + PeFT (PV-Tuning) | 2/8 | 167.05 | 12.83 | 17.44 | 75.74 |
| + SeMI-PV | 2/8 | **213.44** | **6.47** | 7.79 | **92.39** |
| - KAQ | 2/8 | 203.05 | 6.68 | **7.71** | 82.92 |
| + KAQ + LAQ | **1.5/8** | 174.24 | 8.98 | 8.84 | 78.08 |

It may be noted that the AdamW-based PeFT method actually performs substantially better on all metrics than the best PV Tuning-based approach. However, a subjective examination of results using human eye-power (Figure 10 and more examples in Appendix **??**) shows the AdamW-based

approach to perform considerably worse. This discrepancy is unexplained and may point towards issues with the underlying metrics. After all, it is also unusual that both our team and He et al. (2024a) find quantization to extremely low bit-widths to result in FID and sFID scores much *better* than those of the unquantized model.

## 4.5 FLOPs Reduction on ImageNet 256x256

Our key focus in our paper is in the reduction of the RAM or VRAM required for storage of the model weights at inference time, at which we exceed all previous solutions. However, we might also want to reduce the FLOPs required for inference. By default, we may simply decompress the weights from their compressed representation (a very rapid operation) prior to the layer operation. This approach incurs no FLOPs advantage from the weight quantization. Alternatively, we may make use of an efficient inference kernel (Section 3.3).

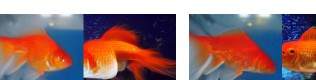

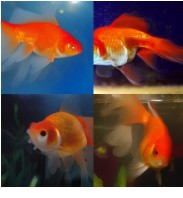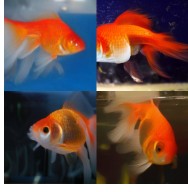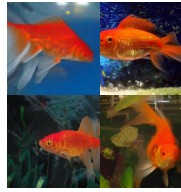

Figure 9: At **W2A8** on LDM-4 ImageNet with $g = 9$ (+KAQ), quantization with SeMI-PV produces outputs considerably more faithful to the original model, despite scoring worse on all ablated metrics.

Owing to the substantial technical investment involved, we have not implemented the efficient inference kernel in a manner which actually accelerates model inference. This is typical for papers on DM quantization, and works such as Li et al. (2023); He et al. (2024a) also make claims regarding BOPs (Bitwise OPeration) or latency without a demonstrated speed-up. However, our method is distinguished by the lack of assumptions about hardware support for small integer arithmetic. We display our results in Table 5.

Table 5: Latency and FLOPs of our method on LDM-4 ImageNet $256 \times 256$. Latency measured for generation of 4 images at 20 inference time-steps using the DDIM sampler. FLOPs measured for a single forward pass on a batch of 4 samples using *fvcore* (AI, 2019).

| Method | Bit-width (W/A) | FLOPs (GFLOPs) | IS↑ | FID↓ | sFID↓ | Precision↑ (%) |
|---|---|---|---|---|---|---|
| FP | 32/32 | 399.52 | 364.73 | 11.28 | 7.70 | 93.66 |
| AQUATIC-Diff + Infer. Kernel | 2/8 | 320.27 (-19.84%) | 213.44 | **6.47** | **7.79** | 92.39 |
| AQUATIC-Diff + Infer. Kernel | 1.5/8 | 255.05 (-36.17%) | 174.24 | 8.98 | 8.84 | 78.08 |

## 5 Conclusion

In this work, we have introduced codebook-based additive vector quantization to diffusion models for the first time. In order to account for the unique features of diffusion models, such as the convolutional U-Net and the progressive denoising process, we have introduced techniques such as Convolutional Kernel-Aware Quantization (KAQ), Layer Heterogeneity-Aware Quantization (LAQ), and Selective Momentum Invalidation PV-Tuning (SeMI-PV). Our method has achieved state-of-the-art results in extremely low-bit quantization. Not only have we set a new Pareto frontier on the LDM-4 benchmark at 20 inference steps, we have also quantized this standard benchmark task to **W1.5A8** for the first time. Additionally, our approach allows for flexibly balancing quantization and inference efficiency and achieves hardware-agnostic FLOPs savings.

**Limitations and future work.** Although AQUATIC-Diff achieves excellent results on a variety of metrics, including some which are state-of-the-art, we are not as efficient in terms of pure GPU hours compared to earlier PTQ+PEFT works such as He et al. (2024a). In part, this results from the slowness of the AQLM layer-wise quantization (Egiazarian et al., 2024) and of the PV-Tuning optimizer (Malinovskii et al., 2024), in comparison to straight-through estimation using Adam (Kingma & Ba, 2015) as applied in He et al. (2024a). In order to address this, work can be invested in the development of faster gradient-based optimization algorithms for additive vector quantization.

## 6 REPRODUCIBILITY STATEMENT

Along with the discussions of methodological procedure and hyperparameter settings in paper, we release our code via an anonymous download link, allowing for the main results to be easily reproduced. Furthermore, in Appendix A.1 we explain the details of the FLOPs-reducing efficient inference kernel and provide proof of its FLOPs reduction. We hope that this provision will be useful to our respected reviewers.

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

# A APPENDIX

## A.1 PROOF OF FLOPS SAVINGS VIA EFFICIENT INFERENCE KERNEL

Consider a convolutional layer with a weight matrix $F$ consisting of $C_{out}$ individual filters $\{F_i\}_{i=1}^{C_{\text{out}}}$, where each filter has dimensions $C_{in} \times h_1 \times w_1$. The forward pass on an input $H$ can be described as the channel-wise concatenation:

$$G = \bigotimes_{i=1}^{C_{out}} H * F_i, \tag{7}$$

where $F \in \mathbb{R}^{C_{out} \times C_{in} \times h_1 \times w_1}$, $F_i \in \mathbb{R}^{C_{in} \times h_1 \times w_1}$, $H \in \mathbb{R}^{C_{in} \times h \times w}$, $H * F_i \in \mathbb{R}^{h \times w}$, and $*$ denotes the convolution operation (non-batched). Note that we have implicitly padded the convolution so as to keep the spatial dimensions the same. We may now apply the classic formula for FLOPs of a non-batched convolution operation:

$$\text{FLOPs} = C_{\text{out}} \times C_{\text{in}} \times h \times w \times h_1 \times w_1 \times 2. \tag{8}$$

Now, instead consider the decompression of a weights matrix quantized via AQLM:

$$\widehat{\mathbf{W}} = \sum_{m=1}^{M} C_{b_{1,m}}^{(m)} \oplus \cdots \oplus \sum_{m=1}^{M} C_{b_{2^{kg/M},m}}^{(m)}, \tag{9}$$

with $\oplus$ as the concatenation operator and $b_{im} \in \mathbb{R}^{2^{kg/M}}$ as the code assigned to the $i$-th group of weights and $m$-th codebook under $k$-bit quantization, where $g$ is the group size and $M$ the number of codebooks. We may think instead of the decompression of a convolutional filter where $g = h_1 \times w_1$:

$$F_i = \sum_{m=1}^{M} C_{b_{1,m}}^{(m)} \oplus \cdots \oplus \sum_{m=1}^{M} C_{b_{2^{kh_1 w_1/M},m}}^{(m)}, \tag{10}$$

with $\oplus$ as instead the stacking operator, so that the tensor dimensions work out. Substitute:

$$G = \bigotimes_{i=1}^{C_{\text{out}}} H * \left( \sum_{m=1}^{M} C_{b_{1,m}}^{(m)} \oplus \cdots \oplus \sum_{m=1}^{M} C_{b_{2^k h_1 w_1 / M}, m}^{(m)} \right). \tag{11}$$

A rearrangement, keeping in mind the manner in which convolution commutes with summation and stacking, grants us:

$$G = \bigotimes_{i=1}^{C_{\text{out}}} \sum_{j=1}^{C_{\text{in}}} \left( \sum_{m=1}^{M} H_j * C_{b_{1,m}}^{(m)} \oplus \cdots \oplus \sum_{m=1}^{M} H_j * C_{b_{2^k h_1 w_1 / M}, m}^{(m)} \right). \tag{12}$$

We may at this point do the tedious work of counting the FLOPs:

$$\begin{aligned} \text{Total FLOPs} = {} & M \times 2^k \times C_{in} \times h \times w \times h_1 \times w_1 \ \text{ multiplications } + \\ & M \times 2^k \times C_{in} \times h \times w \times (h_1 \times w_1 - 1) \ \text{ additions } + \\ & M \times C_{out} \times C_{in} \times h \times w \ \text{ additions}. \end{aligned} \tag{13}$$

Landing us at $C_{in} > M \cdot 2^{9k/m}$ as the breakpoint at which our FLOPs count goes down for a $3 \times 3$ 2-D convolutional kernel.

### A.2 Example of misleading FID result for AdamW vs SeMI-PV (Enlarged).

FP Model · · · · · · · · · · W2A8, SeMI-PV · · W2A8, AdamW

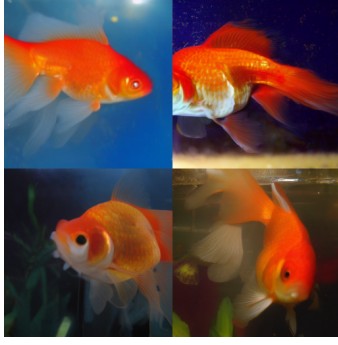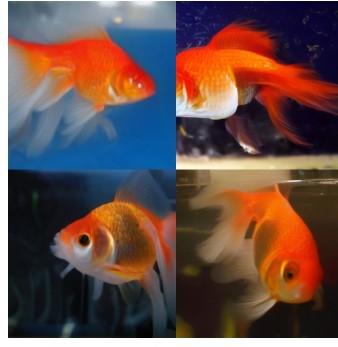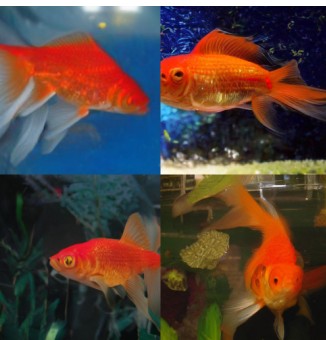

Figure 10: At **W2A8** on LDM-4 ImageNet with $g = 9$ (+KAQ), quantization with SeMI-PV produces outputs considerably more faithful to the original model, despite scoring worse on all ablated metrics.

