# OpenReview forum: "AQUATIC-Diff: Additive Quantization for Truly Tiny Compressed Diffusion Models"
_ICLR.cc/2025/Conference — ICLR 2025 Conference Withdrawn Submission_

### Official Review · Reviewer_Cem8 · 2024-10-24

**Soundness:** 1
**Presentation:** 2
**Contribution:** 1
**Rating:** 3
**Confidence:** 4

**Summary:**

This paper examines codebook-based vector quantization of diffusion models. Th process has two steps. First, each layer is vector quantized using per-layer calibration, minimizing the pre-activation error from the original model. Then, parameter-efficient fine-tuning is performed. The results are compared to previous methods in various metrics.

**Strengths:**

1. The problem being addressed is important.
2. The paper contains code.
3. The method shows good empirical results in Table 2.

**Weaknesses:**

1. The method aims to compress the model, but it is not clear if this translates to any benefit during inference, after accounting for the method overhead (e.g., it has more scales). The only part that seems relevant to this issue is Table 5. Table 5 claims to have latency and FLOPs results, but unfortunately I don't see any latency results. Also, why is the baseline 32/32? Why not 16/16 or 8/8, which should have near 32/32 image quality? From the reported FLOPs I can estimate the proposed method would significantly increase the FLOPs in comparison to such (16/16 and 8/8) baselines, so it is not clear if there is any benefit from using it.

2. The empirical results are not extremely convincing. The best results are shown in Table 2. But even there, the results are not very far from the results of EfficientDM, while both methods seem to have very different costs, so it is hard to know if it is beneficial to use the proposed method. For example, in Table 3, EfficientDM has somewhat worse metrics than the proposed method, while needing roughly 50% less wall time, even though it requires 5 times more fine-tuning steps. Lastly, in Table 1, EfficientDM has better performance than the proposed method, with less GPU time. I understand the authors say EfficientDM code was not available, but as it is published work, I have to believe their reported results in this review.

3. The main FID metrics used for comparisons are problematic. As the authors point out themselves (lines 485-487) these metrics can improve when the visual results look worse. The problem with FID scores was also observed in previous work (e.g., see Appendix E in [SDXL: Improving latent diffusion models for high-resolution image synthesis, ICLR 2024]). This may also explain why the results reported here improve with higher quantization levels. Therefore, the paper could benefit from using additional, more accurate metrics (e.g., PSNR from the image of the original model), and perhaps more generated images to allow for visual inspection. I am aware this is a systemic issue of the literature, which makes it problematic to compare with previous works. However, on the other hand, it is hard for me to make comparisons based on such an unreliable metric (the IS score is also problematic in some cases, as the authors admit [lines 404-405]).

4. The scale of the experiments can be improved, with some experiments on larger scale models that are more relevant for practitioners, such as Stable diffusion (as was done here [TFMQ-DM: Temporal Feature Maintenance Quantization for Diffusion Models, CVPR 2024]).

5. I found section 3 rather dense, and it was hard for me to understand what are the novel parts, and how significant these are in comparison to previous works. Maybe it would be beneficial to extend the explanation a bit more (if there is no room there, then in the Appendix).

Minor points:

1. What is the Precision metric used in the Tables, or the “ID” metric used in Table 1? They are not defined anywhere.
2. Why are two different operators are used to denote concatenation (see eq. 4 vs. eq. 5)?
3. I do not understand what we are supposed to see in Figure 8, and how does this support the proposed method.
4. Notation changes from $b_{i,m}$ in eq. 4 to $b_{im}$ in the line below it.

**Questions:**

See weakness section.

---

### Official Review · Reviewer_9LxF · 2024-10-27

**Soundness:** 2
**Presentation:** 3
**Contribution:** 2
**Rating:** 5
**Confidence:** 4

**Summary:**

In this paper, the authors applied the vector quantization techniques, specifically those proposed in the quantization area of LLMs, to the diffusion models. To facilitate such application, they made several observations and modifications. Empirically I would appreciate the results and using the LLM quantization techniques. However, after reading the paper I feel I did not gain sufficient insights from this paper about what is special about quantization in diffusion models.

**Strengths:**

1. This work utilizes the emerging techniques from recent quantization works.

2. I like how authors report multiple aspects of evaluation, such as algorithm runtime, FIDs, IS, and precision. They are being honest with all metrics.

**Weaknesses:**

* My major concern about this work is it is too incremental from existing works. I am not against combination works (meaning applying method from A to B). However, I expect to have some interesting observations or insights when applying other works into Diffusion models.  For example:
  1. It is intuitive that vector quantization may obtain better performance than uniform quantization. But does the author verify the weight distribution in diffusion models? From the experiments, I did not observe a significant improvement over another uniform quantization baseline. Even more, vector quantization is theoretically a lot slower than uniform W/A quantization, especially on GPUs.
  2. KAQ and LAQ are somehow too straightforward. KAQ lowers the group size to 9 for convolution layers, significantly increasing the codebook memory.
  3. Why do we have to perform PeFT on a convolutional-based architecture? PeFT was originated on LLM with billion-level parameters. Is it possible to perform full model tuning directly on diffusion models? Can we apply PeFT to CNNs post-training quantization?
  4. Other observations including AdamW are a bit engineering. In terms of scientific findings they are really incremental.

* KAQ is way too expensive for de-quantization and inference with this extremely small group size. Even for LLMs the smallest group size is generally 64 (for uniform quantization). I was wondering if EfficientDM applies KAQ-level grouping for quantization parameters. If not, how will your method perform without KAQ?

**Questions:**

See my weakness above.

---

### Official Review · Reviewer_5uZ9 · 2024-11-03

**Soundness:** 3
**Presentation:** 1
**Contribution:** 3
**Rating:** 3
**Confidence:** 4

**Summary:**

In this paper, the authors introduce AQUATIC-Diff, a method for compressing diffusion models with substantial compression ratios. Unlike traditional uniform scalar quantization, AQUATIC-Diff employs vector quantization and leverages quantization-aware fine-tuning to enhance performance. Experiments on both unconditional and image-conditioned diffusion models demonstrate that this approach achieves lossless W4A8 quantization.

**Strengths:**

* The FLOPs reduction achieved by AQUATIC-Diff is noteworthy. Unlike traditional weight-only quantization, which primarily reduces bitwise operations (BOPs), reducing FLOPs can directly decrease latency on off-the-shelf hardware.
* The authors have included their code with the submission to ensure reproducibility.

**Weaknesses:**

* The paper is difficult to follow, with overly long sentences (e.g., in the abstract) and inconsistent citation formatting. Figures are also hard to interpret due to short captions, and there is a missing reference (Line 485).
* The paper's technical contribution and novelty appear limited. It primarily applies vector quantization to diffusion models with quantization-aware fine-tuning. Techniques like Convolutional Kernel-Aware Quantization and Layer Heterogeneity-Aware Quantization (LAQ) seem more like design choices. The fine-tuning approach has already been introduced in EfficientDM.
* Limited experiments: There is no evidence that the proposed method can be applied to recent text-to-image models. All experiments are conducted on unconditional and image-conditioned diffusion models, which are somewhat too simple. Testing on text-to-image models like Stable Diffusion would strengthen the evaluation.
* Insufficient visual results: All visual results presented are of the golden fish class, which is not convincing. The authors should include a more comprehensive set of visual results to demonstrate effectiveness.
* The experiments are conducted solely on U-Net models, and the method is tailored for this architecture. However, the current trend favors Diffusion Transformers (DiT). This raises questions about the practicality and generalizability of the proposed method.

**Questions:**

* Line 53: SDXL was not the state-of-the-art diffusion model at the time of submission; SD3 and the FLUX.1 were already available. Additionally, SDXL’s U-Net has only 2.6B parameters, not 6.6B as stated. Please clarify this discrepancy.
* Figure 7: Why is $\mathbf{x_{t+1}}$ shown as an output of the Optimizer?

---

### Official Review · Reviewer_WGro · 2024-11-05

**Soundness:** 3
**Presentation:** 2
**Contribution:** 3
**Rating:** 5
**Confidence:** 3

**Summary:**

A novel approach for quantizing diffusion models using additive vector quantization

**Strengths:**

The paper applies codebook-based additive vector quantization to diffusion models for the first time, adapting techniques previously used for LLM quantization.

The method achieves unprecedented compression levels, including the first successful W1.5A8 quantization.

The approach allows for a dynamic trade-off between quantization-time GPU hours and inference-time savings, combining benefits of both Post-Training Quantization (PTQ) and Quantization-Aware Training (QAT).

**Weaknesses:**

The paper primarily focuses on the LDM-4 ImageNet model. What about others?

While the paper mentions that the most time-consuming stage is highly parallelizable, it doesn't provide a detailed analysis of the computational requirements for the quantization process compared to existing methods - AWQ, GPTQ, QuaRot [1], TesseraQ [2].

Main concern for me is what is the main contribution here? I feel that such quantization tricks already exist for LLM domain. So what have the authors found or are specifically observing that makes their technique work on diffusion models. Can one of the existing LLM quantization process mentioned above be applied here?

Will there be any issue of overfitting during QAT? And what ll be its impact on stability/convergence?

[1] Ashkboos, Saleh, et al. "Quarot: Outlier-free 4-bit inference in rotated llms." arXiv preprint arXiv:2404.00456 (2024).

[2] Li, Yuhang et al. "TesseraQ: Ultra Low-Bit LLM Post-Training Quantization with Block Reconstruction." arXiv preprint arXiv:2410.19103 (2024).

**Questions:**

See weaknesses

---

### Note · Authors · 2024-11-12

I have read and agree with the venue's withdrawal policy on behalf of myself and my co-authors.